

# Dead sands: bioerosion of alien foraminiferal shells in a Mediterranean seagrass meadow

Martin Vohník[1]

[1]Department of Mycorrhizal Symbioses, Institute of Botany, Czech Academy of Sciences, Průhonice, 25243, Czechia

*Correspondence to*: Martin Vohník (vohnik@ibot.cas.cz)

**Abstract.** Foraminiferans are diverse macroscopic protists abundant in (sub-)tropical seas, often forming characteristic benthic communities known as "living sands". Numerous species have migrated through the Suez Canal to the Mediterranean, some turning invasive and gradually outcompeting the indigenous species. The most expansive *Amphistegina lobifera* often creates thick seabed sediments, thus becoming an important environmental engineer. However, little is known about the turnover of

its shells in the invaded ecosystems. Using vital staining, stereomicroscopy, scanning electron microscopy, cultivation and DNA fingerprinting, I investigated the vital status, destruction/decomposition and mycobiota of *A. lobifera* in the rhizosphere of the dominant Mediterranean seagrass *Posidonia oceanica* in an underwater Maltese meadow (average 284 shells/g, representing 28.5% of dry substrate weight), in comparison with epiphytic specimens and *P. oceanica* roots. While 78% of the epiphytes were alive, nearly all substrate specimens were dead. On average, 80% of the epiphytes were intact, compared to

21% of the substrate specimens. Abiotic dissolution and mechanical damage played only a minor role, but some bioerosion was detected in 18% and >70% of the epiphytic and substrate specimens, respectively. Few bioerosion traces could be attributed to fungi and the majority probably belonged to photoautotrophs. The seagrass roots displayed fungal colonization typical for this species and yielded 81 identified isolates, while the surface-sterilized substrate specimens surprisingly yielded no cultivable fungi, compared to other 16 identified isolates obtained from the epiphytes. While the epiphytes´ mycobiota was

dominated by ascomycetous generalists also known from terrestrial ecosystems (alongside with, e.g., a relative of the "rock-eating" extremophiles), the roots were dominated by the seagrass-specific dark septate endophyte *Posidoniomyces atricolor* and additionally contained a previously unreported lulworthioid mycobiont. In conclusion, at the investigated locality, dead *A. lobifera* shells seem to be regularly bioeroded by endolithic non-fungal organisms, which may counterbalance their accumulation in the seabed substrate.




# 1 Introduction

Foraminiferans (=forams; SAR: Rhizaria: Retaria, see (Irwin et al., 2019)) are amoeboid eukaryotic protists producing large networks of very thin cytoplasmic extrusions (reticulopodia) and living enclosed in genetically fixed single or multichamber

tests (=shells) made of various organic and inorganic materials. With several thousands of recent species, forams represent one of the most diverse groups of marine protist, being found in all marine environments from the tropics to the polar regions, from brackish to hypersaline waters and from the intertidal to the depths of the ocean trenches (e.g., (Pawlowski, 2009); (Altenbach, 2011); (Sabbatini et al., 2014)). Nevertheless, forams are especially abundant in tropical and subtropical seas where their tests form a principal source of calcium carbonate ((Kennett, 1982); (Schiebel, 2002); (Langer, 2008)). Marine forams are both

planktonic and benthic; the latter group is significantly more diverse and encompasses larger symbiont-bearing forams forming specific assemblages known as "living sands" that often dominate tropical and subtropical photic seabed substrates (see (Lee and Anderson, 1991) and references therein).

        Many (sub-)tropical foraminiferal species have been introduced from the Red Sea through the Suez Canal to the comparably colder Mediterranean Sea (following the so-called Lessepsian route, see (Galil, 2006)), including several larger

forams, and while some of them are rather rare, others became important benthic components dominating local foram communities and profoundly changing the structure and type of the invaded habitats, thus acting as "environmental engineers" ((Zenetos et al., 2008); (Yokeş and Meriç, 2009)). Arguably the most abundant alien foram in the Mediterranean Sea is the calcareous symbiont-bearing *Amphistegina lobifera* (Rotaliida: Amphisteginidae, Fig. 1a-c; (Weinmann et al., 2013)). It is widely distributed in the Eastern Mediterranean Basin ((Koukousioura et al., 2010); (Yokes et al., 2014)) and thanks to its high

dispersal potential aided by increasing water temperatures ((Guy-Haim et al., 2017); (Prazeres et al., 2020)), it gradually expands westwards, the current distribution limit laying between the coast of southern Tunisia, the Maltese Islands and the Adriatic coast along southern Albania ((Yokes et al., 2007); (Langer and Mouanga, 2016); (El Kateb et al., 2018)). In the Levantine Basin, it often forms very dense populations resulting in seabed sediments up to 80 cm thick that in a way resemble the tropical living sands ((Yokes et al., 2014); Figs 1d and 2a).

While foram ecology, evolution, physiology and taxonomy have attracted significant research attention, comparably less is known about the post-mortem fate of their shells, or more specifically, about the agents causing foram shell degradation/destruction during early burial (cf. (Martin, 1999)). The main abiotic processes (disaggregation, corrosion/dissolution, fragmentation, mechanical abrasion, etc.) have been studied to a larger degree (e.g., (Berger, 1967); (Denne and Sen Gupta, 1989); (Kotler et al., 1992); (Berkeley et al., 2009)) and it is evident that they have profound selective

effects on dead foram assemblages. For example, abiotic dissolution especially affects forams with smaller and calcareous tests, thus significantly modifying the composition of the foram sediment/palaeoecological record ("taphonomic bias", e.g., (Martin and Wright, 1988); (Green et al., 1993); (Murray and Alve, 1999); (Nguyen et al., 2009)). In contrast, the biotic processes (bioerosion, decomposition) have been studied to a lesser extent and mostly at the descriptive level (e.g., (Kloos, 1982); (Nielsen and Nielsen, 2001); (Malumián et al., 2007); (Cherchi et al., 2012); (Frozza et al., 2020)), despite that they





may cause impacts similar to the abiotic ones (cf. (Perkins and Halsey, 1971)) and, for example, alleviate the negative impact of the accumulation of alien foram shells in the invaded ecosystems (cf. (Yokeş and Meriç, 2009)).

Bioerosion can be defined as the destruction and removal of consolidated substrates (lithic and plant/woody) by the action of organisms ((Neumann, 1966); (Bromley, 1992); (Tribollet et al., 2011)) while decomposition as the breaking down of dead organic matter by the action of (micro-)organisms (Kothe, 2011). Bioerosion can be divided into bioabrasion (caused

by various grazers), biocorrosion (chemical attack) and boring (various macro- or microborers) ((Neumann, 2008); for alternative definitions see (Bromley, 1992) and (Tribollet et al., 2011)). Macro- and microborers constitute the endolithic guild of bioeroders, in general represented by soft bodied organisms producing shallow stationary borings in hard substrates ((Golubic et al., 1981); (Tapanila, 2008)). Microborers comprise extremely small sponges, bryozoans and especially algae, cyanobacteria and fungi and from the ecophysiological perspective, they can be divided into autotrophs and heterotrophs

((Bromley, 1992); (Tapanila, 2008)). With a few exceptions (like fungi seeking and utilizing organic skeletal matrix and subsequently resting in the resulting borings, see (Warme, 1975)), boring activities are typically connected with creating a living space/shelter in a hard substrate (Schönberg and Wisshak, 2014) while during decomposition, the respective organisms obtain food (i.e., source of carbon, energy, etc.) from various organic substrates. For historical and practical reasons, bioerosion and decomposition have been typically studied by different research communities (palaeontologists and biogeologists vs.

biologists of different specializations) that use different methodological approaches (for bioerosion, see (Golubić et al., 1970); (Hirsch et al., 1995); (Wisshak and Tapanila, 2008); (Golubic et al., 2019); (Heřmanová et al., 2020) and many others).

Fungi commonly colonize both abiotic and biotic (both living and dead) substrates and arguably represent the most understudied group of marine bioeroders, despite that they are known from practically all marine habitats (e.g., (Golubić et al., 2005); (Gadd, 2011); (Amend et al., 2019)). While they may be the dominant microborers in the aphotic zone, they are also

quite common in shallower depths where they colonize various biotic substrates like carapaces of crustaceans, shells of molluscs, submerged driftwood, thalli of calcareous algae, etc. (e.g., (Kohlmeyer, 1969); (Kohlmeyer et al., 2004); (Golubić et al., 1975); (Rämä et al., 2014) and many others). Fungal interactions with forams are not very well understood, despite that the foram biomass may represent a potentially important trophic resource in many marine ecosystems (cf. (Lipps, 1983); (Lee and Anderson, 1991)). The available literature is scarce and most of the studies are observational, without an evidenced

explanation of the nature of the observed interaction. For example, under laboratory conditions, some unidentified fungi were observed to colonize and possibly also bioerode shells of *Archaias angulatus* (Miliolida: Soritidae) (Butcher and Steinker, 1979). Some ascomycetous arenicolous species can colonize dead tropical forams and produce sporocarps inside and on the surface of their shells while under laboratory conditions, the shell material may serve as a sole source of nutrients for the fruiting fungi ((Kohlmeyer, 1984, 1985)(Kohlmeyer, 1985); (Volkmann-Kohlmeyer and Kohlmeyer, 1993); also see Fig. 1A

in (Tokura, 1983)). Finally, (Shroba, 1993) ascribed some taphonomic features observed on the shells of temperate benthic forams to fungi, but without a detailed documentation and identification of the responsible microborers.

From the foram point of view, fungi are generally not considered as a part of their diet (cf. (Lee and Anderson, 1991)). However, (Langer and Gehring, 1993) proposed that certain small motile epiphytic species that produce organic traces





consisting of sulphated glycosaminoglycans might do so to farm bacteria and fungi for subsequent consumption. In addition,
in the intracellular content of some intertidal benthic forams investigated by (Chronopoulou et al., 2019), there was a high
relative abundance of fungal DNA (belonging to the members of Saccharomycetes and Exobasidiomycetes), suggesting some
kind of a potential trophic interaction. While it is difficult to imagine that forams could extracellularly digest or graze intact
living mycelium, they might feed on the often very minute fungal spores and/or bacteria living in the hyphosphere, as proposed
for some soil testate amoebae (Vohník et al., 2009, 2011).

105        Fungi are not only decomposers/saprobes, but also engage in various symbiotic interactions along the mutualistic-
parasitic continuum. In the Mediterranean context, a rather curious fungal symbiosis is that with the roots of the dominant
seagrass *Posidonia oceanica* (Alismatales: Posidoniaceae). While the first detailed observations upon the root anatomy of the
seagrass had been published ca. 130 years ago (Sauvageau, 1889), the symbiosis was discovered only recently (Vohník et al.,
2015). However, since the discovery, it has been reported from every single investigated site in the NW Mediterranean Sea
(Borovec and Vohník, 2018; Vohník et al., 2016, 2017). It is formed by a single ascomycetous mycobiont not known from any
other host or environment that was very recently described as *Posidoniomyces atricolor* (Pleosporales: Aigialaceae) (Vohník
et al., 2019). Despite its apparent omnipresence in the whole northern Mediterranean Sea (personal observation) and the fact
that it morphologically resembles the dark septate endophytic association ubiquitous in the roots of the majority of the
terrestrial plants (e.g., (Lukešová et al., 2015)), next to nothing is known about its functioning as well as significance for both
the mycobiont and the host seagrass. Nevertheless, besides the dominant *P. atricolor*, some other fungi associate with *P.
oceanica* roots, including *Corollospora maritima* (Microascales: Halosphaeriaceae) (Cuomo et al., 1985), an ascomycete found
to form sporocarps on the shells of *Amphistegina* sp. from Hawaii (see Fig. 1 in (Kohlmeyer, 1985)).

        In January 2017, during a search for the phytomyxid colonizing another Lessepsian migrant from the Red Sea, the
alien seagrass *Halophila stipulacea* (Alismatales: Hydrocharitaceae, see (Kolátková et al., 2020)), I encountered an abundant
*A. lobifera* population at Balluta Bay, St. Julian´s, Malta. At some places, its numerous shells formed layers many centimetres
thick, evoking a Mediterranean version of the tropical living sands (Fig. 2a). While I had not found any *H. stipulacea*, the site
was occupied by vigorous patches of *P. oceanica* whose leaves often protruded from the seabed substrate full of *A. lobifera*
(Fig. 2b).

        At places of their high abundance, (Butcher and Steinker, 1979) encouraged studies of factors contributing to foram
bioerosion, because an understanding of the mechanisms of diagenesis of their shells would significantly contribute to
interpretation of the history of carbonate depositional environments. In addition, (Kohlmeyer, 1985) suggested that
representatives of the genus *Amphistegina* might be good sources of recent "higher" marine fungi (that colonize and bioerode
their shells). Hence, I returned to the same place in May 2018, collected samples of *A. lobifera* shells from the rhizosphere of
the seagrass (and epiphytic specimens + the seagrass roots for comparison) and investigated them using various approaches
(vital staining, stereomicroscopy, light and scanning electron microscopy, fungal isolation and DNA fingerprinting): first, to
assess the vital status of the *A. lobifera* specimens as well as their frequency in the substrate and second, to address two central
questions of this study, i.e., 1/ what is the fate of dead *A. lobifera* shells in the *P. oceanica* rhizosphere and 2/ whether the



fungi inhabiting the seagrass roots colonize the dead shells, thus contributing to their bioerosion. Since the seagrass roots are tightly coupled with a unique spectrum of marine fungi (see above), I hypothesized that these would be the primary bioeroders
of dead *A. lobifera* shells.

## 2 Materials and methods

### 2.1 Sampling

Epiphytic specimens of *Amphistegina lobifera*, rhizosphere substrate and roots of the seagrass *Posidonia oceanica* were collected using scuba diving at three different microsites (ca. 10 m apart) at a depth of ca. 6 m at Balluta Bay, St. Julian´s,
Malta (GPS: N35.915685, E14.495578) on 28th May 2018. The epiphytic specimens were collected from *P. oceanica* leaves and seaweeds growing in the immediate vicinity of the seagrass (Fig. 2c, d) and the substrate containing *A. lobifera* specimens (volume ca. 50 ml) from the seagrass rhizosphere. All samples were divided in two sub-samples of equal volume, one for (stereo-)microscopic screening and one for mycobiont isolation, and processed as described below.

### 2.2 Screening of *Amphistegina* shells and *Posidonia* roots

The sub-samples containing *A. lobifera* shells were further divided into halves; one half was stained for two weeks with rose Bengal, washed repeatedly with tap water and dried to distinguish alive and dead specimens (Walton, 1952) while the other half was dried and used for counting (to establish the abundance of *A. lobifera* specimens in 1 g of the dried substrate), weighting (the total weight of *A. lobifera* specimens in 1 g of the dried substrate), measuring (the diameter of the substrate specimens) and (stereo-)microscopy (to document bioerosion/colonization, dissolution and mechanical damage of the
epiphytic + substrate specimens). To measure the diameter of the substrate specimens, random 100 mg of substrate shells per each microsite were separated and the measurements were performed on all shells occurring in three separate fields of view using an Olympus SZX12 stereomicroscope (magnification 12.5×) and the QuickPHOTO MICRO ver. 3.2 software (Promicra, Czechia).

To document bioerosion/colonization, the respective shells were first roughly screened using the stereomicroscope
and subsequently, 30 random shells per type and microsite were assessed using a FEI Quanta 200 ESEM scanning electron microscope (FEI Company, USA) in the Low Vacuum mode at room temperature (detailed SEM screening is a lengthy process so the total number of screened shells was primarily limited by the working time available at the SEM microscope). With respect to bioerosion/dissolution, they were sorted out into six qualitative categories, i.e., 1/ intact (=not affected, Fig. 1), 2/ non-bioeroded but partially dissolved, 3/ bioeroded and partially dissolved, 4/ only bioeroded – low level, 5/ only bioeroded –
intermediate level and 6/ only bioeroded – high level (of bioerosion). Additionally, surface colonization by macroepiphytes and mechanical damage were recorded (independently of the former six categories) (for illustration see Fig. 3). I did not attempt to determine the respective microborers taxonomically; instead, they were conservatively distinguished into two classes, i.e., fungi and non-fungal organisms. Because the traditional sorting based on the diameter of the borings (e.g., (Perkins and Halsey,





1971)) is not very reliable (see (Golubić et al., 1975)), the borings were assigned to the former class only when intact hyphae

were first observed on the shell surface using a stereomicroscope (for illustration see Fig. 4).

Random *P. oceanica* root segments from each microsite were screened for fungal colonization using a compound Olympus BX60 microscope at high magnifications (400× and 1000×) as detailed in (Vohník et al., 2015). In brief, the fine terminal roots were separated from the root system, washed with tap water, their transversal and longitudinal semi-thin sections were prepared using a razor blade and these were mounted in lactoglycerol in glass slides and evaluated for fungal colonization

using the compound microscope.

Stereomicroscopy and light microscopy photographs were taken with an Olympus DP70 camera, the Deep Focus Mode embedded in QuickPHOTO MICRO ver. 3.2 was employed when needed. The obtained photos were modified for clarity and contrast as needed and assembled into Figures using Paint.net ver. 4.0.13 (dotPDN LLC, Rick Brewster and contributors).

### 2.3 Mycobiont isolation and identification

The protocol for isolation and identification of fungi colonizing *A. lobifera* shells and *P. oceanica* terminal roots comprised methods identical to those described in more detail in (Vohník, 2020); this paper also describes their rationale and intuitive troubleshooting. In brief, the low-carbon potato carrot agar (PCA) used for mycobiont isolation was prepared by boiling 40 g of carrots and 40 g of potatoes separately in 500 ml of deionized water for 5 min. The resulting broth was autoclaved at 121°C for 20 min, diluted 1:1 with sterile deionized water, supplemented with agar (10 g/l; HiMedia, India), again autoclaved at

121°C for 20 min and when cooled but still liquid, it was supplemented with Novobiocin sodium salt (50 mg/l; Sigma-Aldrich, Germany) to prevent growth of bacteria. The medium was poured into plastic square 25-compartment Petri dishes and left to solidify under UV light overnight.

50 epiphytic and 50 substrate shells and 50 root segments (ca. 3–4 mm long) were selected randomly from the samples from all three microsites. The shells and the root segments were surface-sterilized 30 s in 10% SAVO (common household

185 bleach; Unilever, Czechia; 100% SAVO contains 47 g kg$^{-1}$, i.e., 4.7% sodium hypochlorite = NaClO), 3x washed with sterile deionized water and then transferred onto the surface of the solidified medium in the dishes. Additionally, 25 substrate shells from one microsite were not surface-sterilized but only serially washed with sterile deionized water and then treated as above, serving as a control treatment. The isolations took place during the day of collection. Petri dishes with the shells and root segments were incubated at room temperature in the dark and periodically checked for fungal growth. After six months, all

190 visible fungal cultures were counted, assigned codes and identified as detailed below. As *Posidoniomyces atricolor*, the dominant root mycobiont of *Posidonia oceanica*, is notoriously slow-growing (Vohník et al., 2019), the dishes were re-examined after another five months and all new cultures were counted, assigned codes and identified as detailed below.

For mycobiont molecular identification, total DNA was extracted from all fungal cultures producing enough mycelium using an Extract-N-Amp Plant Kit (Sigma-Aldrich, Germany) following manufacturer's instructions. The ITS1-

195 5.8S-ITS2 region (ITS) of the nuclear ribosomal DNA (nrDNA) was amplified using the ITS1F + ITS4 primer pair and the partial large subunit (LSU) nrDNA of some isolates was amplified using the LR0R + LR7 primer pair. The PCR and gel





electrophoresis parameters were the same as in (Vohník et al., 2016). The PCR products were purified and sequenced in the Macrogen Europe Laboratory (Macrogen Europe, The Netherlands) using the ITS1, ITS4, LR0R and LR7 primers.

The obtained sequences were screened in Finch TV v1.4.0 (https://digitalworldbiology.com/FinchTV) for possible machine errors and manually edited/trimmed. Where available, the reverse sequences (i.e., those obtained with the ITS4 and LR7 primers) were converted to reverse complement sequences and aligned with the corresponding forward sequences, yielding consensus sequences (contigs) representing the respective fungal isolates. The resulting ITS sequences were subsequently subjected to BLAST searches in GenBank and those not belonging to *Posidoniomyces atricolor* were aligned in ClustalW (Thompson et al., 1994) implemented in BioEdit v7.2.5 (Hall, 1999). The resulting alignment was used as a matrix for a neighbour joining (NJ) analysis (default settings) in TOPALi v2.5 (Biomathematics & Statistics Scotland, www.topali.org) to delimit molecular operational taxonomic units (MOTUs); the threshold limit for grouping of sequences was set at 99%. One MOTU (#14) was delimited based on the only available LSU sequences. Sequences within separate MOTUs were further aligned to screen their heterogeneity and their taxonomic position was checked using Blast Tree View (NJ, default settings). Fungal taxonomy follows the MycoBank Database (http://www.mycobank.org/, accessed during June-October 2020).

# 3 Results

## 3.1 Screening of *Amphistegina* shells and *Posidonia* roots

On average, 78.1% of the epiphytic *A. lobifera* specimens were alive (averages for the three microsites: 53.7, 83.3 and 97.4%). In contrast, a great majority (>99%) of the substrate specimens from all three microsites were dead. On average, there were 284 specimens in one gram of the dried substrate (395, 282 and 175 shells), representing on average 28.5% of the total weight of the dried substrate (43.1, 26.5 and 15.8%). The average diameter of the substrate shells was 1.32 ± 0.23 mm (mean + SD; min. 0.52, max. 2.08 mm).

On average, 80% of the epiphytic *A. lobifera* shells were intact (i.e., showed no signs of biotic or abiotic degradation), compared to only 21% of the substrate shells. Only abiotic dissolution was observed in just a few shells (2% and 8%, respectively). Whereas some degree of bioerosion was observed on average only in 18% of the epiphytic shells, it was >70% in the case of the substrate shells. Highly bioeroded were on average 3% of the epiphytic shells, compared to 13% of the substrate shells (for details see Table 1, for examples see Fig. 3). Only a minor part of the bioerosion traces could be unambiguously attributed to fungi, typically only in a combination of stereomicroscopy followed by SEM (Fig. 4).

All screened root segments displayed the dark septate endophytic colonization typical for *P. oceanica* collected in the NW Mediterranean Sea (Fig. 5a) that has been documented in terms of morphology, anatomy and ultrastructure in several recent papers (see above).





## 3.2 Mycobiont isolation and identification

In total, 107 fungal isolates were obtained from the 150 surface-sterilized *P. oceanica* root segments (86 isolates; ca. 57% isolation success) and the 150 epiphytic (19; ca. 13%) and the 25 non-sterilized substrate (2; ca. 8%) shells of *A. lobifera*. The

150 surface-sterilized *A. lobifera* shells yielded no isolate. Out of these, 97 were identified with the aid of molecular fingerprinting (Table 2) and they belonged to 14 distinct MOTUs (Table 3). While the epiphytic shells yielded 12 MOTUs that were mostly represented by a single isolate (max. two), the root segments yielded two other MOTUs represented by 67 and 14 morphologically distinct isolates (Fig. 5b). There were no overlaps between the shell- and root-associated MOTUs (Table 3). The epiphytic shell mycobiota comprised generalists like *Alternatia*, *Cladosporium* and *Penicillium* spp., known also from

terrestrial ecosystems, alongside with one isolate probably representing a new species in the genus *Knufia* and four MOTUs that could be reliably identified only at the class level. The root mycobiota was at all three microsites dominated by *P. atricolor* whose compact blackish slow growing colonies (Fig. 5b, c) appeared to develop from intraradical (micro-)sclerotia (Fig. 5d). However, at one microsite, the root segments also yielded a previously unreported lulworthioid mycobiont (MOTU 14) probably representing a new species in the Lulworthiales (Table 3).

## 4 Discussion

This study took place at the current NW distribution limit of the alien foram *Amphistegina lobifera* in the Western Basin of the Mediterranean Sea, yet the abundance of its shells in the seabed substrate was comparable with or even exceeded those reported from the comparably warmer Eastern Basin (average 28.5% reported here vs. 32.7% reported from the Antalya coast in Turkey, see (Yokes et al., 2014); max. 395 shells/g reported here vs. max. 178 shells/g reported from the Israeli coast, see

(Hyams et al., 2002). Thus, despite that the thickness of the substrate containing *A. lobifera* shells by far did not reach the impressive 60–80 cm reported by (Yokeş and Meriç, 2009), the alien foram shells did represent a significant part of the bottom sediment at the investigated Maltese locality and profoundly changed the seabed character (i.e., from calcareous rocks combined with mineral sand and pebbles to a homogenous layer with a large proportion formed by the biogenic calcareous matter, see Fig. 2a, b). Interestingly, in contrast to the tropical "living sands", practically all *A. lobifera* substrate specimens

were dead. On the other hand, similar has been reported, e.g., for substrate shells from Key Largo, Florida, USA (Martin and Wright, 1988). The seagrass *Posidonia oceanica* is known to produce "matte", i.e., an important seabed sediment composed of siliciclastic and biogenic carbonated materials mixed in various ratios with organic matter (mainly *P. oceanica* roots, rhizomes and leaves) that can be several meters thick and thousands of years old (e.g., (Serrano et al., 2012)). From the geobiological point of view, it would be interesting to investigate how the matte formation is influenced by the accumulation

of dead *A. lobifera* shells in the seabed substrate.

Investigations of the processes beyond the foram shell breakdown and turnover are important not only because of the information loss and taphonomic bias inherent to the transition from living to dead foram assemblages (e.g.,(Martin and Wright, 1988)), but also for a better understanding of the factors limiting the accumulation of alien foram shells in the invaded



ecosystems, e.g., through abiotic dissolution (e.g., (Green et al., 1993)) and bioerosion (e.g., (Cherchi et al., 2012)),
transformation of the shells into lime mud, i.e., the important matrix of both recent and ancient calcareous sediments (e.g.,
(Debenay et al., 1999)), etc. Here, while the abiotic dissolution and mechanical damage contributed only little, the majority
(>70%) of the substrate shells showed at least some signs of bioerosion, with 13% being highly bioeroded. This is opposite to,
e.g., the findings of (Berkeley et al., 2009) who investigated tropical intertidal sediments in north Queensland, Australia and
concluded that the calcareous test degradation during early burial was primarily driven by dissolution, not bioerosion.
However, the reason(-s) for this difference remain unknown. Nevertheless, the data gathered here suggest that bioerosion may,
at least to a certain degree, counterbalance the accumulation of alien foram shells in the seabed and thus alleviate the negative
impact of the alien foram environmental engineering ((Zenetos et al., 2008); (Yokeş and Meriç, 2009)).

Surprisingly, a great majority of the bioerosion traces seemed to belong to non-fungal organisms (probably
cyanobacteria and/or microscopic algae). Congruently, and in contrast to the main hypotheses, not only the substrate shells did
not share any fungi with the *Posidonia oceanica* roots, they did not yield any cultivable fungi at all. This is an unexpected
result, because cultivable fungi are ubiquitous in marine ecosystems and regularly colonize calcareous substrates including
foram shells (cf. (Kohlmeyer, 1969, 1984, 1985)). In addition, the epiphytic shells were colonized by fungal ubiquitous
generalists as well as specialists and the seagrass roots were regularly colonized by specific symbiotic fungi, including a
member of the Lulworthiales that comprise common marine ascomycetes, some of them colonizing foram shells (see
(Kohlmeyer et al., 2000)). Nevertheless, a few substrate shells did display apparent signs of fungal colonization by dark septate
hyphae (Fig. 4) that actually resembled the mycelium of the dominant *P. oceanica* root mycobiont (see below). However, an
attempt to clone fungal DNA from such shells ended with inconclusive results (data not shown).

The disappearance of cultivable fungi from the substrate shells observed in this study is difficult to explain and one
can only speculate about its reasons. For example, since most of the substrate specimens were dead, the respective shells were
presumably empty, i.e., without sufficient organic matter to support the fungal growth. However, many marine ascomycetes
are notoriously slow-growing (i.e., they need little nutrients), including the dominant *P. oceanica* root mycobiont (see (Vohník
et al., 2019) and references therein) and, e.g., all the foram-associated tropical marine fungi reported by (Kohlmeyer, 1984,
1985) probably developed on and/or within dead shells. A more likely explanation is allelopathy, a phenomenon common also
among marine microorganisms (see (Hellio et al., 2000); (Gross, 2003); (Cepas et al., 2019) and many others). Here, the
antagonists could be the (presumably autotrophic) microbioeroders abundant in the substrate shells and/or the fungi inhabiting
*P. oceanica* roots. Indeed, while numerous cultivable fungi have been recently obtained from nearly all *P. oceanica* tissues,
they were absent in the apical parts of the leaves that, however, commonly displayed colonization by microscopic
algae/cyanobacteria (B. Soperová and M. Vohník, unpublished results). In addition, while it is still unknown, e.g., how far can
reach the mycelium of *P. oceanica* root-symbiotic fungi, it is interesting to note that their diversity is, at least in the NW
Mediterranean Sea, extremely low and dominated by a single mycobiont (Vohník et al., 2016, 2017, 2019). While data from
other seagrasses are too few to allow any robust comparisons, such dominance is extremely rare both in freshwater aquatic



and terrestrial ecosystems (e.g., (Vandenkoornhuyse et al., 2002)) and may suggest some kind of antagonism between the dominant root mycobiont and other marine fungi.

Also in this study, the seagrass roots were dominated by *P. atricolor*, a pleosporalean fungus not known from any
other hosts or environments, and the microscopic observations presented here (Fig. 5c, d) provide further indirect evidence that this mycobiont is responsible for the root colonization pattern ubiquitous in the NW Mediterranean Sea (Fig. 5a, see (Vohník et al., 2015)). The seagrass roots additionally yielded a hitherto unknown lulworthioid mycobiont and the epiphytic shells an isolate with affinities to the genus *Knufia* that comprises highly destructive extremotolerant lithobionts that, e.g., often bioerode Mediterranean historical monuments exposed to outdoor conditions (see (Isola et al., 2016) and references
therein). While these isolates represent interesting and potentially important mycobionts and illustrate how little we know about the diversity of marine fungi (see (Gareth Jones, 2011) and references therein), their more detailed taxonomic assignment remained outside the scope and dimensions of this study.

## 5 Conclusions

In the first study focused on the fate of *A. lobifera* during early burial in an invaded ecosystem, I found out that practically all
its substrate specimens were dead and regularly bioeroded by presumably photoautotrophic microborers, not marine fungi. Their taxonomic affinities as well as possible antagonistic interactions with the latter remain unknown and beg further investigations. In contrast, the epiphytic *A. lobifera* specimens yielded a relatively diverse spectrum of mycobionts, at least in comparison with the roots of the seagrass *P. oceanica*, which comprised both ubiquitous generalist and specialist well-adapted to bioerode calcareous substrates. The switch from fungi in the epiphytic shells to non-fungal organisms in the substrate shells
is curious and deserves elucidation, possibly through a study focusing on allelopathic interactions between these two microborer guilds. Nevertheless, a few substrate shells were indeed colonized by unidentified fungus/fungi with dark mycelium and possible future studies on interactions of forams with fungi may consider focusing on foram specimens more intimately associated with seagrass roots.

*Data availability.* The nrDNA sequences obtained in this study (see Table 3) are publicly available in GenBank at NCBI (MT636935-41, MT636972-84).

*Author contribution.* MV designed the study, organized and performed the sampling on Malta and fungal isolation and identification, analysed the data and wrote the paper.

*Competing interests.* The author declares that he has no conflict of interest.



*Acknowledgments.* This study was supported by the Czech Science Foundation (GAČR 18-05935S) and the Institute of Botany, Czech Academy of Sciences (RVO 67985939). I wish to thank Viktorie Kolátková for introducing me to the world of forams,
the assistance during sampling on Malta and the help with assessing the vital status and abundance of *A. lobifera* in the samples investigated in this study. I also thank Zuzana Heřmanová for sharing the micro-CT image (Fig. 1c), Jiří Machač for taking the SEM photos and assembling the figures and Katarína Holcová for leading the Czech Science Foundation project #18-05935S "From past to present: fossil vs. recent marine shelled organisms as a substrate for colonization and bioerosion".

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





**Figure 1: The invasive foraminiferan *Amphistegina lobifera* from Balluta Bay, St. Julian´s, Malta.**

(a) ventral view, SEM, bar = 200 µm; (b) dorsal view, SEM, bar = 200 µm; (c) micro-CT 3D reconstruction (coloured to contrast inner structure), bar = 300 µm; (d) magnified view of an *A. lobifera* assemblage (some specimens stained with rose Bengal), bar = 500 µm. The shells collected by Martin Vohník, photos taken by Jiří Machač, Institute of Botany, Czech Academy of Sciences, Průhonice under Martin Vohník's supervision (a, b), Zuzana Heřmanová, National Museum, Prague (c) and Martin Vohník (d).






**Figure 2: Substrate and epiphytic communities of *Amphistegina lobifera* investigated in this study.**

(a) in situ view of the investigated seabed substrate containing numerous *A. lobifera* specimens; (b) leaves of the seagrass
*Posidonia oceanica* being buried in the substrate containing numerous *A. lobifera* specimens; (c), (d) epiphytic specimens of
*A. lobifera* occurring on the leaves of *P. oceanica* (arrows) and the surrounding seaweeds. All photos taken by Martin Vohník.


**Figure 3: Examples of bioerosion, abiotic dissolution and epiphytic colonization of shells of the invasive foraminiferan *Amphistegina lobifera* visualized by scanning electron photography.**

*Amphistegina lobifera* substrate shells displaying low (a), medium (b, c) and high (d, e, f) levels of bioerosion (cf. Table 1); (g), (h), (i) typical bioerosion traces found in the substrate shells; (j – n) various degrees of abiotic dissolution, sometimes combined with bioerosion (j, m, n); (o), (p) examples of epiphytes on *A. lobifera* shells (arrows). Bars 300 μm (a, b, f, j), 200 μm (c, d, e, k – p), 75 μm (i), 50 μm (g) and 25 μm (h). The shells collected by Martin Vohník, all photos taken by Jiří Machač under Martin Vohník's supervision.





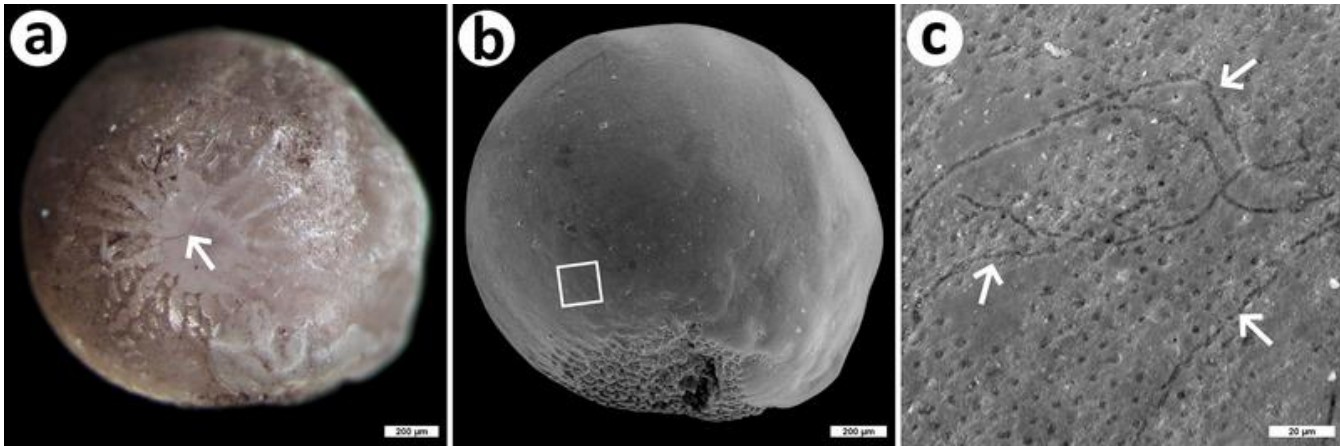


**Figure 4: An example of fungal colonization of a shell of the invasive foraminiferan *Amphistegina lobifera*.**

(a) dorsal view of an *A. lobifera* shell apparently colonized by dark brown mycelium (arrow), stereomicroscopy, bar = 200 µm; (b) ventral view of the same shell as in (a), SEM, bar = 200 µm, the square delimits the area magnified in (c) and displaying fungal traces on the surface (arrows), SEM, bar = 20 µm. The shell collected by Martin Vohník, photos taken by Martin Vohník

(a) and Jiří Machač under Martin Vohník's supervision (b, c).









**Figure 5: Colonization pattern and root mycobionts of the dominant Mediterranean seagrass *Posidonia oceanica*.**

(a) the typical colonization pattern in the seagrass roots that resembles the ubiquitous terrestrial dark septate endophytes; (b) 25-compartment plastic Petri dish filled with nutrient medium and with fungal colonies emerging from some of the surface-sterilized seagrass root segments. Note that one morphotype produces diffuse substrate mycelium (it corresponds to MOTU #14 = the Lulworthiales sp. MV-2018, see Table

3), while the other remains small and limited to the surface of the root segments or their immediate vicinity (MOTU #13 = *Posidoniomyces atricolor*). In this particular case, 9 root segments did not yield any fungal mycelium, i.e., the isolation success reached 64%. (c) detail of



compact colonies of *P. atricolor* emerging from a surface-sterilized root segment, SEM, bar = 200 µm; (d) longitudinal section through a root segment yielding a compact colony of *P. atricolor*, note that the surface mycelium originates from an enlarged intraradical sclerotium (arrow), SEM, bar = 200 µm. All photos taken by Martin Vohník.





**Table 1: Bioerosion, abiotic dissolution, macroepiphytic colonization and mechanical damage of _Amphistegina lobifera_ shells in numbers.**

Epiphytic and substrate shells of the invasive foraminiferan _A. lobifera_ were collected at three different microsites at a depth of ca. 6 m at Balluta Bay, St. Julian´s, Malta, for details on their investigation see Materials and Methods.

| Microsite # | Type of shells (n = 30) | Categories of shell bioerosion/abiotic dissolution | | | | | | Shells with macro-epiphytes | Mechanically damaged shells |
|---|---|---|---|---|---|---|---|---|---|
| | | 1: Intact (not affected) | 2: Only partially dissolved | 3: Bioeroded + partially dissolved | 4: Only bioeroded - low | 5: Only bioeroded - medium | 6: Only bioeroded - high | | |
| 1 | Epiphytic | 20 | 0 | 1 | 4 | 3 | 2 | 1 | 2 |
| 1 | Substrate | 8 | 3 | 5 | 8 | 2 | 4 | 1 | 5 |
| 2 | Epiphytic | 27 | 1 | 1 | 0 | 0 | 1 | 1 | 0 |
| 2 | Substrate | 5 | 2 | 4 | 6 | 6 | 7 | 2 | 3 |
| 3 | Epiphytic | 25 | 1 | 0 | 2 | 2 | 0 | 1 | 0 |
| 3 | Substrate | 6 | 2 | 5 | 14 | 2 | 1 | 2 | 0 |
| Average (all sites) | Epiphytic | 80% | 2% | 2% | 7% | 6% | 3% | 4% | 2% |
| Average (all sites) | Substrate | 21% | 8% | 16% | 31% | 11% | 13% | 3% | 9% |


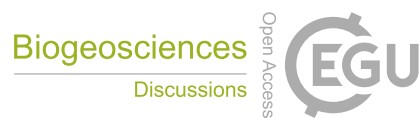

**Table 2: Results of mycobiont isolation from shells of the foram *Amphistegina lobifera* and roots of the seagrass *Posidonia oceanica*.**
The mycobionts were isolated into pure cultures and the Molecular Operational Taxonomic Units (MOTUs) were delimited as described in Materials and Methods.

[1] data only for non-sterilized substrate shells (surface-sterilized substrate shells yielded no isolate)

| Sample type | Microsite #1 | | Microsite #2 | | Microsite #3 | |
|---|---|---|---|---|---|---|
| *A. lobifera* epiphytic shells | **8 isolates** (6 identified) | **6 MOTUs** (#1, 5, 6, 8, 9, 11) | **4 isolates** (all identified) | **3 MOTUs** (#1, 7, 12) | **7 isolates** (6 identified) | **4 MOTUs** (# 2, 3, 4, 10) |
| *A. lobifera* substrate shells[1] | **0 isolates** | - | **2 isolates** (0 identified) | - | **0 isolates** | - |
| *P. oceanica* root segments | **19 isolates** (17 identified) | **1 MOTU** (# 13) | **29 isolates** (all identified) | **1 MOTU** (# 13) | **38 isolates** (35 identified) | **2 MOTUs** (# 13, 14) |



**Biogeosciences** Open Access
Discussions
EGU

**Table 3: Taxonomic affinities of the Molecular Operational Taxonomic Units representing mycobionts isolated in this study.**
Taxonomic affinities (identity) of the delimited Molecular Operational Taxonomic Units (MOTUs) are based on comparing representative ITS and/or LSU nrDNA sequences with those available in a public database as described in Materials and Methods.

[1] EPI = epiphytic shells of the foram *Amphistegina lobifera*, POS = terminal roots of the seagrass *Posidonia oceanica*. The numbers after the codes (i.e., 1, 2 and 3) represent the three sampling microsites (see Table 1).

| MOTU # | MOTU occurrence (no. of identified isolates)[1] | MOTU identity | Reference isolates [sequences in GenBank] |
|---|---|---|---|
| 1 | EPI-1 (1), EPI-2 (2) | *Penicillium* sp. MV-2018A | MLT-5 [MT636974 (ITS)] |
| 2 | EPI-3 (2) | *Penicillium* sp. MV-2018B | MLT-56 [MT636983 (ITS)] |
| 3 | EPI-3 (1) | *Penicillium* sp. MV-2018C | MLT-55 [MT636982 (ITS)] |
| 4 | EPI-3 (2) | *Penicillium* sp. MV-2018D | MLT-51 [MT636980 (ITS)] |
| 5 | EPI-1 (1) | *Knufia* sp. MV-2018 | MLT-8 [MT636977 (ITS), MT636937 (LSU)] |
| 6 | EPI-1 (1) | Dothideomycetes sp. MV-2018A | MLT-7 [MT636976 (ITS)] |
| 7 | EPI-2 (1) | *Alternaria* sp. MV-2018 | MLT-28 [MT636979 (ITS), MT636939 (LSU)] |
| 8 | EPI-1 (1) | Dothideomycetes sp. MV-2018B | MLT-4 [MT636973 (ITS), MT636936 (LSU)] |
| 9 | EPI-1 (1) | Sordariomycetes sp. MV-2018A | MLT-6 [MT636975 (ITS)] |
| 10 | EPI-3 (1) | Sordariomycetes sp. MV-2018B | MLT-52 [MT636981 (ITS), MT636940 (LSU)] |
| 11 | EPI-1 (1) | *Cladosporium* sp. MV-2018A | MLT-3 [MT636972 (ITS), MT636935 (LSU)] |
| 12 | EPI-2 (1) | *Cladosporium* sp. MV-2018B | MLT-27 [MT636978 (ITS), MT636938 (LSU)] |
| 13 | POS-1 (17), POS-2 (29), POS-3 (21) | *Posidoniomyces atricolor* | MLT-87 [MT636984 (ITS)] |
| 14 | POS-3 (14) | Lulworthiales sp. MV-2018 | MLT-72 [MT636941 (LSU)] |