# Peer review of "Bioerosion and fungal colonization of the invasive foraminiferan *Amphistegina lobifera* in a Mediterranean seagrass meadow"

_Biogeosciences, 2020_

## Referee Comment (RC1)

[referee-annotated manuscript omitted]

---

## Referee Comment (RC2)

[referee-annotated manuscript omitted]

---

## Author Response (AR1)

**Response to R1:**

This is an interesting and novel manuscript and reports on the vital status, destruction/decomposition and mycobiota communities of A. lobifera in the rhizosphere and on ephiphytic shells from the Mediterranean seagrass Posidonia oceanica. The novel aspects concern the study of the mycobiota on living and dead shells of the foraminifer Amphistegina lobifera (as epiphytes and as dead shells) and within the rrizosphere of Posidonia oceanica. While the analysis of seagrass roots yielded 81 identified isolates, the surface-sterilized substrate specimens revealed no cultivable fungi. Only 16 identified isolates were obtained from the epiphytes.

The manuscript is well written and provides new insight into the fate, destruction and bioerosion of foraminiferal shells.

Three sites were investigated, where shells of the epiphytic symbiont-bearing foraminifera live on the seagrass and eventually accumulate in the sand. The sediments were found to eventually accumulate dead shells of Amphistegina, but the shells do not (yet) accumulate as thick layers, as has been reported from other sites in the eastern Mediterranean Sea.

*Authors Response (AR): thank you – being a novice in the foram field, I am glad that what I did in this study makes sense/is interesting not only from the mycological perspective.*

As such, I find that the title of the ms does not reflect the content of this paper for the 3 reasons outlined below:
1. The focus of this study is on mycobiota communities and bioerosion
2. The Amphistegina rich deposits do not form thick sands (yet), as reported from other sites in the eastern Mediterranean Sea
3. The sands are not "dead", as they contain abundant other living organisms including living foraminifera (but not studied here).

*AR: true – I changed the title of the ms to "Bioerosion and fungal colonization of the invasive foraminiferan Amphistegina lobifera in a Mediterranean seagrass meadow"*

**Technical issues concerning the sampling methods:**
The sampling procedure for the collection of epiphytes is not well described and as such it is difficult to replicate this study (how many leaves were collected, how were the epiphytes collected? Collection of the epiphytes by placing a bag over the leaves or by just cutting the leaves makes a big quantitative difference. A clarification of this issue is needed.

*AR: true – I added a more detailed explanation and the respective part of the ms now reads as follows:*

*"The epiphytic specimens originated from P. oceanica leaves and seaweeds growing in the immediate vicinity of the seagrass (mostly Dictyota dichotoma) (Fig. 2c, d). The former was in situ scraped off the surface of the leaves using opened 50 ml plastic test tubes, the latter was individually collected with tweezers from the seaweed surface in the laboratory and both were eventually pooled (no attempt was made to calculate an exact seagrass : seaweed ratio but the majority of the epiphytic shells were from seaweeds). To obtain the rhizosphere substrate, P. oceanica rhizomes with intact healthy-*

*looking leaves were gently lifted up a little and the substrate right below was collected into opened 50 ml plastic test tubes*
*with seawater."*
The material analyzed includes not only leaves of Posidonia oceanic but also other seaweeds growing in the immediate
vicinity. What are the other seaweeds? Epiphytic foraminifera communities may differ substantially when you collect them
from different types of algae and seagrasses (see e.g. Langer 1993, Epiphytic foraminifera or papers by Kitazato).
*AR: most if not all were D. dichotoma (see above). I did not focus on the total foram community but only on A. lobifera*
*(that anyway represented the great majority of the epiphytes recognizable with the naked eye) with special emphasis on*
*its substrate shells. The epiphytes were in a way a control treatment and I had expected a quite opposite result, i.e., the*
*epiphytes (nearly) free of fungi vs. the substrate shells full of fungi (overlapping with those from the seagrass roots). After*
*reading your comment (and given that the epiphytes yielded some interesting fungal isolates), I realize I should have been*
*more precise, i.e., keeping + investigating the seagrass and the seaweed epiphytes separately. An inspiration for future!*
The references concerning the invasion of alien/invasive species of foraminifera, environmental engineers, carbonate
production of tropical foraminifera are often "second hand" references and do not cite the original source/relevant papers.
I have added numerous comments in the marked-up manuscript and suggested additional references.
*AR: perhaps true (being a novice in the field, I cannot really tell) and thank you for the many suggested alternative*
*references, I will factor them into a revised version of my ms.*
Other than this, I find this paper to be of interest to a wide range of readers and recommend publication with
minor/moderate revisions.
*AR: thank you. It is an interdisciplinary research and I hope it will be interesting not only for*
*microbiologists/mycologists/marine ecologists etc. but also for the foram people.*
Attached is my marked-up manuscript.
*AR: thank you – I incorporated your suggestions into the revised MS*
Martin Langer
Please also note the supplement to this comment:
https://bg.copernicus.org/preprints/bg-2020-452/bg-2020-452-RC1-supplement.pdf
*AR: thank you for your time and the fitting comments! Martin Vohník*

**Response to R2:**

A very interesting contribution on the taphonomy of a dead assemblage of Amphisteginids next to Malta Island.

There is a very promising and result rich study on diversity and presence / abundance of fungal activity recorded on both the rhizosphere and the benthic community.

*AR: thank you – I indeed had a feeling that the available literature on foram vs. fungi interactions is very scarce.*

I was surprised that quite a deal of emphasis was given on possible substrate reduction by the activity of bioeroders / dissolution effects and not a word is spent on possible transportation effects.

*AR: here the emphasis was on the effect of biological processes (colonization/bioerosion) and their possible consequences so I guess it does make sense that both Introduction and Discussion focus on these issues (papers studying effects of transport typically do not spend many words on, e.g., bioerosion or colonization by fungi). In fact, I also had factored in abiotic processes like dissolution/mechanical damage (the latter probably closely connected with transport), but they seemed to play only a minor role. On the other hand, it is true that some literature on transport should have been mentioned – it will be done in the revised version of this ms (in the paragraph around the lines 55-59 in the original manuscript, as you pointed out in the supplement to your review)*

At 6 meters water depth hydrodynamics can be massive and Posidonia meadows can act as shields for all those particles that are transported within the meadow to get trapped and accumulated.

*AR: definitely true, that´s the reason I included a paragraph mentioning this issue in Discussion (lines 251-255 in the original manuscript – I guess it belongs there rather than to Introduction because I did not investigate this process, it is a suggestion for future studies for anyone interested).*

A dense Amphistegina made substrate of severals tens of cm thick (less than 60 is specified, but I did not see a specific number) can be the result of accumulation by mass transport.

*AR: definitely agreed – but once again, I did not study these processes nor am I familiar with them for the study area so I can say very little… No specific number was provided for the thickness as it had not been rigorously measured but some rough estimation is provided in the lines 120-121 of the original manuscript.*

very minor details, got me the feeling that the author is not an expert on larger foraminifera, the word foraminiferan is often used insted of the classic foraminifera, Bengal Rose is used to check for living specimens when in larger forams the best method is by looking at the very distinctive symbiont colouration after few hours of rest after sampling.

*AR: Indeed I am a novice in this very interesting field – just based on literature searches it seemed to me that both terms are used with very similar frequency, foraminiferan being a younger common/trivial name while foraminifera older/classic. I decided for the former (in fact I mostly use just "foram/-s") mainly because in the latter case, some authors*

*use Foraminifera (which is a taxonomic unit) while others foraminifera (which I consider a common name), a situation*
*confusing for me.*
*Thank you for suggesting an alternative method for distinguishing the living foram specimens, will compare it with Bengal*
*Rose next time! (since most of the substrate specimens were scored as dead I think the possible differences would be*
*minimal in the case of this study)*
I should not judge the grammar and the syntax as I am not a native speaker but, in my opinion, the text is written is a very
good english, clear and sound. Structure of the MS is appropriate and the references are lacking all those regarding transport,
that, to my opinion, is a critical issue here.
*AR: I will certainly factor in some references regarding transport at appropriate places (also as suggested in the*
*supplement of your review) in a revised version of this ms.*
A number of markups are directly on the attached PDF
*AR: thank you – they were incorporated in the revised MS!*
regards
Antonino Briguglio
Please also note the supplement to this comment:
https://bg.copernicus.org/preprints/bg-2020-452/bg-2020-452-RC2-supplement.pdf
*AR: thank you for your time and the fitting comments/suggestions! Martin Vohník*